# Living Longer or Better—Patient’s Choice in Cardiac Surgery Is Gender-Dependent—A Multicenter Study

**DOI:** 10.3390/jcm12247596

**Published:** 2023-12-09

**Authors:** Britt Hofmann, Epp Rae, Ulrike Puvogel, Mihaela Spatarelu, Salah A. Mohamed, Almoan Bungaran, Sebastian Arzt, Magdalena L. Laux, Klaus Matschke, Richard Feyrer, Hans-Hinrich Sievers, Ivar Friedrich, Bernd Niemann, Rolf-Edgar Silber, Andreas Wienke, Andreas Simm

**Affiliations:** 1Department of Cardiac Surgery, Mid-German Heart Center, University Hospital Halle (Saale), 06120 Halle (Saale), Germany; britt.hofmann@uk-halle.de (B.H.);; 2Department of Cardiac and Vascular Surgery, 35392 Gießen, Germany; 3Department of Cardiac Surgery, 54292 Trier, Germany; 4Department of Cardiac and Thoracic Surgery, 23538 Lübeck, Germany; 5Department of Cardiac Surgery, 91054 Erlangen, Germany; 6Department of Cardiac Surgery, 01307 Dresden, Germany; 7Department of Cardiovascular Surgery, Heart Center Brandenburg, University Hospital Brandenburg Medical School, Faculty of Health Sciences Brandenburg, 16321 Bernau, Germany; 8Institute of Medical Epidemiology, Biostatistics, and Informatics, Medical Faculty, Martin-Luther-University Halle-Wittenberg, 06112 Halle (Saale), Germany

**Keywords:** patient’s preferences, quality of life, lifespan, gender

## Abstract

In view of the increasing age of cardiac surgery patients, questions arise about the expected postoperative quality of life and the hoped-for prolonged life expectancy. Little is known so far about how these, respectively, are weighted by the patients concerned. This study aims to obtain information on the patients’ preferences. Between 2015 and 2017, data were analyzed from 1349 consecutive patients undergoing cardiac surgery at seven heart centers in Germany. Baseline data regarding the patient’s situation as well as a questionnaire regarding quality of life versus lifespan were taken preoperatively. Patients were divided by age into four groups: below 60, 60–70, 70–80, and above 80 years. As a result, when asked to decide between quality of life and length of life, about 60% of the male patients opted for quality of life, independent of their age. On the other hand, female patients’ preference for quality of life increased significantly with age, from 51% in the group below sixty to 76% in the group above eighty years. This finding suggests that female patients adapt their preferences with age, whereas male patients do not. This should impact further the treatment decisions of elderly patients in cardiac surgery within a shared decision-making process.

## 1. Introduction

Cardiovascular diseases (CVDs) are the most common non-communicable diseases globally, taking an estimated 17.9 million lives yearly [1]. Most of these deaths are among people older than 65 years. Forty years ago, heart operations for patients aged 70 and older or in patients bearing a high risk of complications due to increased frailty were avoided due to the increased risk of perioperative complications. Today, most of the patients within a cardiac surgery department are older than 70. Operative procedures target complex morbidities and combinations of ischemic, degenerative, and functional pathologies of the heart. Nearly all relevant risk factors—such as left ventricular ejection fraction below 30%, critical preoperative condition, heart failure NYHA (New York Heart Association) IV, renal failure, reoperations, cardiac arrhythmias, and pulmonary hypertension—are increased in the older group significantly compared to a younger group [2]. Procedural adoptions have met the increasing risk profile of the patients, and surgeons have successfully reduced the overall mortality. Indeed, compared to the total population, even in elderly patients, average life expectancy can be restored after successful surgery [2]. However, different therapeutic strategies may focus on better long-term outcomes on the one hand or improved short-term recovery on the other hand. Unfortunately, this information about the impact of a treatment option on postoperative quality of life or lifespan is often missing for the patient when making a decision. Therefore, patients should clearly state what they expect from the operation. While it can be assumed among young patients that the desire for a high quality of life and life extension is pronounced, this balance can shift in old age. With increasing age, postoperative quality of life can be more critical for these patients. However, knowledge about the patient’s outcome preferences (longer lifespan versus higher quality of life) is sparse in cardiac surgery, and data mainly originate from partner disciplines. Regarding the treatment of angina, 83% of the patients would “accept any treatment, no matter how extreme, to return to health” [3]. The severity of symptoms may have a deep impact on the patient’s expectations regarding the treatment goals and the choice of treatment options. While patients suffering from heart failure decompensation predominantly desire symptomatic alleviation, recompensated patients on the other hand might additionally focus on the extension of life expectancy [4]. A questionnaire-based study of 662 patients who attended health centers in Spain showed that the willingness to pay for a year with a high quality of life was also influenced by higher education and the income of the respondents [5]. Cancer patients are also at risk of high mortality, and more studies exist regarding the quality of life (QoL) and lifespan (LoL). The weighing up of QoL and LoL often touches on much more circumscribed periods of time, may be perceived as more threatening, and is therefore discussed much more intensively and for longer than in cardiovascular medicine. In contrast, it is often believed that cardiovascular diseases can be treated accordingly for a long time without disadvantages (for example, high blood pressure) and therefore they seem to be less perilous. This is problematic, as data clearly show that heart failure, for example, has a survival rate comparable to aggressive tumors. A survival study of 283,048 patients with heart failure hospitalization has shown that about 48% survive by 3 years, about 34% by 5 years, and about 17% by 10 years. The median survival is 2.8 years [6]. In contrast to patients who often do not experience heart failure as a life-limiting disease, patients with tumors accept the severity of the disease. In a study of 459 patients with advanced cancer, 55% equally valued QoL and LoL, with 27% preferring QoL. This preference was stronger with older age and male gender [7]. This can change if patients come to an end-of-life situation. In a cross-sectional face-to-face survey with 182 seriously ill cancer inpatients in Australia, most (76%) of the patients preferred end-of-life care to focus on quality of life (e.g., mitigating pain and discomfort) even if it meant not living longer [8]. In a study in China with terminal cancer patients, in addition to offering moderate life extension, it was found that improving quality of life during end-of-life care deserved more attention [9]. In another study, which concerned voting on hypothetical cancer cases, more than 20% of cancer patients voted for a treatment that prolonged survival regardless of QoL, whereas only about 2% of healthy oncology healthcare professionals preferred this option. Maximizing QoL at the cost of lifetime was acceptable for 34% of laypersons, 23% of healthcare professionals, but only 15% of patients. Thus, in cancer patients, survival dominates the therapeutic goals in contrast to healthy individuals [10]. As a result, the severity of symptoms, as well as individual morbidity, age, gender, and role as patient or practitioner, influence therapy goals. Therefore, it is necessary to be aware that the optimal cardiovascular therapeutic concept from the practitioner’s perspective does not necessarily correspond to the wishes and values of the (elderly) patients. Moreover, studies on a possible decision between QoL and LoL are missing in cardiac surgery, even though there is a general belief on the part of heart surgeons that QoL becomes more important for elderly patients.

## 2. Methods

Study Participants: our goal was to enroll in this prospective study more than 1000 patients within seven departments for cardiac surgery in Germany, i.e., the Clinic for Cardiac Surgery of the University Hospital Halle (Saale), the Clinic for Cardiac and Vascular Surgery of the University Hospital Gießen, the Clinic for Cardiac Surgery of the Hospital of the Brothers of Charity in Trier, the Clinic for Cardiac and Thoracic Surgery of the University Hospital Lübeck, the Clinic for Cardiac Surgery of the University Hospital Erlangen, the Clinic for Cardiac Surgery of the University Hospital Dresden, and the Clinic for Cardiac Surgery of the Heart Center Brandenburg in Bernau. The study protocol was approved by the local ethical boards of all seven hospitals. All participants provided written informed consent. Inclusion criteria were elective cardiac surgery and age above 18. Exclusion criteria were emergency operations and dementia of the patients. On the one hand, patients with dementia usually undergo cardiac interventions instead of cardiac operations and, therefore, were not within our patient collective. On the other hand, within the pre-treatment consultation, the interviewer tested the patient’s understanding of the consultation/treatment to exclude severe dementia. The following patient data were recorded: age (years); Euro-Score (%); STS Score; sex; body mass index; familial situation (alone, family, nursing home); smoking (yes, no, ex-smoker); syncope (yes, no); myocardial infarction (yes, no); hypertension (yes, no); diabetes (yes, no); lipid disorder (yes, no); chronic obstructive pulmonary disease (COPD, yes, no); peripheral artery disease > II (PAD, yes, no); kidney failure (defined as kidney damage that had existed for three months or longer or a glomerular filtration rate < 60 mL/min/1.73 m^2^; a patient was considered to be ill if they had a history of chronic renal insufficiency or dialysis therapy; no, compensated, dialysis); atrial fibrillation (sinus rhythm, paroxysomal atrial fibrillation, persistent atrial fibrillation); Dyspnea New York Heart Association (NYHA) classification score (no limitations, limitations only at ordinary physical activity, limitations at light physical work, limitations at rest); Canadian Cardiovascular Society grading of angina pectoris (CCS score; no, only at ordinary physical activity, at light physical work, at rest); and reason for operation: aortic valve (no vitium, stenosis, insufficiency, combination), mitral valve (no vitium, stenosis, insufficiency, combination), tricuspit valve (no vitium, stenosis, insufficiency, combination), coronary artery disease (CAD; one, two or three vessel disease). Single question regarding the outcome expectation: What are the important goals to reach with an operation, better quality of life (one to six (one is most important, six least important)) or long life (one to six (one is most important, six least important))? The following central question regarded the patient’s choice for the operation: Regarding the possible outcome of the operation, if you have to choose one option, for which of the two options would you decide: a longer lifespan with reduced quality of life or a shorter lifespan with optimal quality of life? One thousand three hundred forty-nine patients were enrolled. The study was registered at the German Clinical Trials Register (DRKS00025218). The patients’ characteristics are shown in Table 1.

### Statistical Analysis

The sample size was based on feasibility and not determined via a confirmatory sample size calculation. To analyze the influence of age and gender on the preference of the patients with respect to long life vs. quality of life, logistic regression was used. Results are presented as odds ratios (OR) with 95% confidence intervals (95% CI) and *p*-values. Adjusted logistic regression analyses check the impact of multimorbidity and other variables on the observed pattern. Multimorbidity is defined as the presence of more than 2 out of the following 7 conditions: atrial fibrillation, syncope, diabetes, lipometabolic disorder, COPD, PAD, and renal failure.

## 3. Results

To analyze the preference of patients regarding their personal aim of the heart operation, the study participants had to answer before the operation the following questions:

What are the main goals you want to achieve with the upcoming surgery? If the patients were allowed to choose both options—better quality of life and long life—they voted equally for both as essential goals. This decision did not change with age when the data set was analyzed for the whole group as well as for gender-specific groups (male/female). This picture did change if the patients had to decide between two options. For this, they had to answer the following question: Regarding the possible outcome of the operation, if you have to choose one option, for which of the two options would you decide: a longer lifespan with reduced quality of life or a shorter lifespan with optimal quality of life? For analysis, patients were divided into four age groups: below 60, 60–70, 70–80, and above 80. To analyze the influence of age and gender on the preference of patients with respect to long life vs. quality of life, logistic regression was used. Because of the different results for males and females, respectively, data were analyzed together and stratified by gender. Results are presented as odds ratios in favor of QoL (OR) with 95% confidence intervals (95% CI) and *p*-values. In the total group (N = 1349), we see a non-significant increase of the QoL option with age (at the age above 80: odds ratio 1.48, *p*-value 0.05). After dividing the participants into both genders, it is clear that male patients did not change their decision with age. On the other side, females did adapt their answers to their age (Table 2). With increasing age, they chose significantly quality of life as an operational preference by a significant margin (at the age above 80: odds ratio 3.32, *p*-value < 0.01).

Figure 1 shows the proportion of patients within the four age groups choosing the QoL option. It is clearly seen that female patients change their minds, whereas male patients do not. Until the age of 70, female and male patients’ decisions are readily comparable. Above this age, females opt more for the quality-of-life option, whereas male patients stick to their choices. This is somewhat unexpected as, in tumor patients, the quality of life preference is stronger with older age and male gender [7].

A possible alternative explanation of the results could be the effect of morbidity on the decision-making process. In principle, males and females may have different types of morbidities, and this can be more important than sex alone. To clarify this, morbidity was considered within the multivariate regression analysis. Table 3 shows the analysis of patients’ preferences with respect to long life vs. quality of life by gender and age groups, adjusted for multimorbidity. In comparison to Table 2, it appears that adjustment for multimorbidity does not alter the observed pattern with respect to gender and age.

This also holds for analyses adjusted for the single co-morbidities mentioned in the Methods section as well as for the familial situation of the patients (results not shown).

## 4. Discussion

Cardiovascular diseases, a major cause of global mortality, are age-dependent and caused by several degenerative processes. It is discussed that fundamental biological aging mechanisms, summarized as pillars or hallmarks of aging, are responsible for these diseases. López-Otin et al. proposed 12 hallmarks/processes of aging: genomic instability, telomere attrition, epigenetic alterations, loss of proteostasis, disabled macroautophagy, deregulated nutrient-sensing, mitochondrial dysfunction, cellular senescence, stem cell exhaustion, altered intercellular communication, chronic inflammation, and dysbiosis [11]. In addition, oxidative and glycative stress are particularly important in the cardiovascular system as sources of macromolecular damage. These age-associated processes reduce the resilience and resistance of the heart and vessels to stress, ultimately leading to functional impairments and diseases [12]. Besides biological processes, sociological (social networks, support, education, financial resources) and psychological (emotional factors, self-efficacy, anxiety, stress, etc.) developments throughout life will also help to decide between healthy aging and disease development [13]. Due to all the different mechanisms mentioned above during aging, and because of an increase in expected lifespan at birth, more people now reach old age with degenerative diseases needing interventions and surgery, especially in oncology and cardiology/cardiovascular surgery. There are increasing concerns regarding decision-making capacities at old age [14]. Here, diseases like dementia/cognitive impairments are in focus, as they can negatively impact shared decision making as well as the ability to develop and communicate one’s own preferences. There are different abilities needed for decision making: understanding one’s own health condition, reasoning about the consequences of a decision (comparing risks and benefits of options), and communicating one’s own choice. Due to these problems, we decided to exclude patients with dementia from this study. On the other hand, especially in old age, additional factors, such as poor finances, loneliness, and time horizons, are important. In 2022, Boyle et al. developed a new conceptual model for decision making based on cognitive abilities, contextual factors, time horizons, and psychosocial factors [15]. In a study on gender-specific preferences within elective total joint replacement surgery, it was shown that men chose surgery earlier in their disease than women did, whereas women preferred to suffer pain rather than risk surgery [16]. In line with this study, Bookwala et al. showed in a study on older adults that men preferred life-prolonging procedures, compared to women, for treatments like surgery or in response to specific health scenarios [17]. Our results also indicate that especially at old age, women prefer quality of life over life span extension, compared to men.

The assumption of a healthy and fit person with regard to the needs of a sick or frail patient is highly subjective and probably only partially agrees with the treatment goals and risk avoidance. Ahman and Kendell therefore asked rhetorically, “Clearly, surviving cardiac surgery is very important—but is survival the top priority for the 92-year old after bypass surgery who becomes unable to live independently again and whose quality of life is insufferable? Should quality of life be the main factor driving therapeutic decisions for the frail and elderly?” [18]. This predicts that during aging, the priority—in the patient’s opinion—regarding the outcome of a heart operation may change. In principle, cardiac surgery can improve quality of life of patients above 80 undergoing even non-elective (urgent or emergency) operations. Although this surgery is related to high in-hospital mortality, quality of life as well as functional parameters equals that of elderly in the normal population [19]. One of the greatest deficits in the therapy of elderly or decrepit cardiac surgical patients remains that there has not yet been any targeted, prospective independent data collection with regard to the therapy requirements in this patient group. In most cases, when asked the simple question, “Is life span important for you?”, patients will answer in the affirmative. The same is true if they are asked the question about quality of life. This observation is in line with our results. Only if patients have to decide between these two outcome options may one see changes. Our data clearly show gender-specific differences regarding the decision between these two options in terms of the major outcome wishes of quality of life versus length of life. A possible alternative explanation for this result is that male and female patients have different underlying diseases/co-morbidities, which may be more important for the decision-making process. We tested this option and cannot support this idea. The observed gender and age pattern is stable over different adjusted analyses and, therefore, not influenced by single co-morbidities, multimorbidity, or familial situation of the patient.

What can be the cause of this gender difference? In neuroscience, neurobiological differences between men and women (dimorphisms) are perceived in many neurological and psychiatric conditions, making sex an important factor [20]. In cardiovascular diseases, female sex is associated with less mortality risk due to coronary heart diseases and stroke at a young age across economically, socially, and culturally diverse countries [21]. Hormones like estrogens are known to play a cardioprotective physiological role and are thought to be the primary cause of sex-gender differences. Estrogens (E2) exert, for example, a positive influence on hypertension as well as on pathologic heart hypertrophy [22]. Within our study, the preferences regarding quality of life are not different between both sexes until the age of 65. If females become older—far later than the menopause—they decide differently than males. This change is indeed later than the decrease in hormone/estrogenes concentrations, which is mostly finished at the age of 60. Therefore, hormones seem not to be the reason for the observed differences between male and female decisions.

Sex difference in cardiac interventional procedures is a critical issue. One argument is that the underrepresentation of the female sex in cardiac interventional procedures is mainly due to a perceived greater frailty of the female sex. A recent analysis on sex differences on leadless pacemaker implantation showed that men and women have the same adverse event risk in the operative and long-term setting, rebutting this argument that females are a high-risk group of patients within interventional procedures [23]. An argument for sex differences could be that the surgeons (mostly men) argue/explain risks and benefits of an operation differently to male (same sex) in comparison to females (different sex).

To analyze “simulated” real-life decision making, the Iowa Gambling Task (IGT) can be used. Whereas women and men do not differ in terms of choices for safe or risky options per se, it was reported that women may choose the safe option more often after a previous loss than men [24]. In a study on sex-related differences in neural activity during risk-associated decision making (based on the Risky Gains Task), the authors show that men and women call upon very different neural processes and mental resources. They discuss the possibility that these sex-related differences are at least partly imprinted during human evolution [25].

Genetic components can also be an interesting factor, such as the Val158Met polymorphism of the COMT gene. This gene encodes catechol-O-methyl-transferase, an enzyme that breaks down dopamine. Using the IGT performance in a healthy adult sample, Costa et al. showed that COMT can be a genetic marker underlying sex differences in decision making [26].

Another explanation could be the differing real-life situations of elderly men and women. Whereas in Germany in 2006, about 60% of 80-year-old men were married, more than 50% of women were widows at the age of 70 [27]. As women live longer, older women in Europe were more than twice as likely to live alone compared to their male counterparts (United Nations, Department of Economic and Social Affairs, Population Division (2017). World Population Ageing 2017—Highlights (ST/ESA/SER.A/397)). For this reason, quality of life may be more important for older women than for men. Our data suggest that maintaining independence is more central than longevity for senior women. On the other hand, our study does not support the effects of living alone, as the familial situation of the patients did not impact decision making.

To our knowledge, this is the first study demonstrating that female patients adapt their outcome preferences with age whereas male patients do not. The personal preferences of our senior patients should be addressed with an individualized treatment option plan within a shared decision making (SDM) process. In a review on shared decision making in the heart team, it was noted, “Despite recommendations in professional guidelines, SDM is not yet common practice in the Heart Team decision-making process in the treatment of patients” [28].

Some limitations of the study should be noted. Our study population was a population with a low to medium risk profile. This could have had an impact on personal outcome preferences. In addition, the study was focused on co-morbidities that are more traditional. We did not include psychiatric disorders, such as depression or anxiety disorders, in our questionnaire. Therefore, we cannot exclude an impact of these factors on decision making.

## 5. Conclusions

This multi-center study represents the first and sufficiently powered study within cardiac surgery on patient’s preferences regarding the possible goal of an operation. Cardiac operations can increase lifespan up to the level of the normal population and/or increase the quality of life. Physicians do believe that due to the increase in multi-morbidity with age, quality of life will be become the major focus of the patients independent of sex of the elderly. Whereas simple questions—how important is the one or the other specific positive outcome–do not help to distinguish between these both options (better quality of life versus increase in lifespan), a forced decision between two options clarifies the preference. In our study, female patients decide in directions to quality of life with increasing age, whereas male patients do not change their mind. Although the proportion of females living at old age (and becoming patients) is increasing in comparison to males, medicine is still focused on males. Most of the procedures used at the moment are developed by male surgeons treating male patients. In addition, many of the new clinical studies still concentrate on mortality. In light of our results, future studies should focus on female medicine and quality of life as the primary outcome. All overall, we believe that our data reflect senior cardiac surgery patient outcome preferences and open the field for more studies as well as for more individualized treatments.

## Figures and Tables

**Figure 1 jcm-12-07596-f001:**
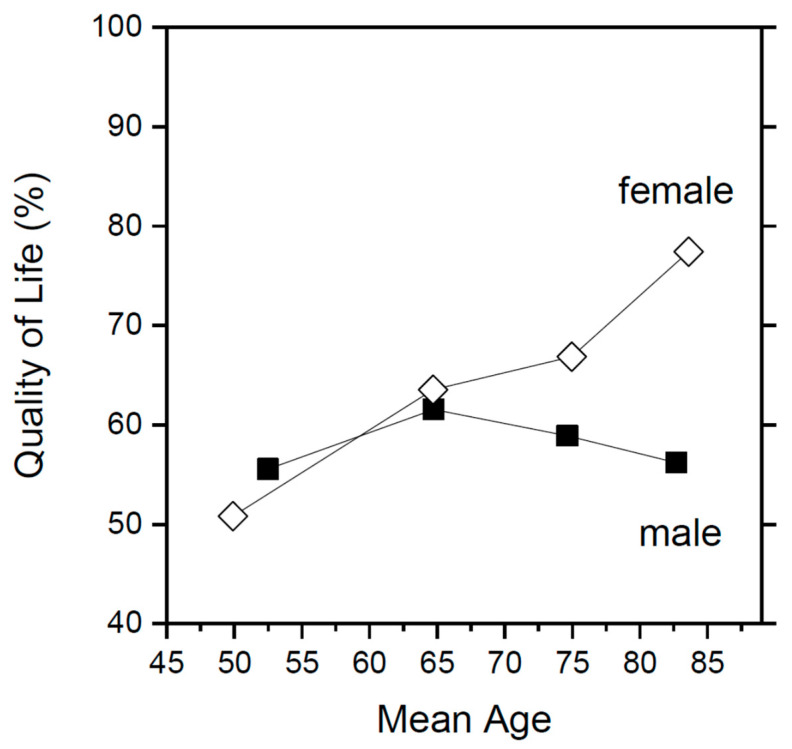
Gender- and age-dependent preferences of cardiac surgery patients.

**Table 1 jcm-12-07596-t001:** Patients’ characteristics.

**Patients (N)**	1349
**Age (years)**	68.49 ± 10.74
**Female (%)**	29.28
**BMI**	28.62 ± 5.24
**EuroScore II (%)**	3.71 ± 5.19
**STS Score (%)**	2.16 ± 2.87
**NYHA Score**	2.65 ± 0.79
**Angina Pectoris—CCSS**	2.00 ± 1.02
**Diabetes (%)**	35.80
**Hypertension (%)**	89.02
**Current smoker (%)**	17.36
**Lipid disorder (%)**	64.76

**Table 2 jcm-12-07596-t002:** Patients’ preferences with respect to long life vs. quality of life by gender and age groups.

	Males (*n* = 954)			Females (*n* = 395)			Total (*n* = 1349)		
	OR	95% CI	*p*-Value	OR	95% CI	*p*-Value	OR	95% CI	*p*-Value
Age below 60	1.00			1.00			1.00		
Age 60–69	1.27	0.88–1.83	0.21	1.69	0.88–3.24	0.12	1.36	0.99–1.87	0.06
Age 70–79	1.14	0.80–1.62	0.46	1.97	1.09–3.56	0.03	1.34	0.99–1.80	0.06
Age above 80	1.02	0.63–1.64	0.95	3.32	1.52–7.23	<0.01	1.48	0.99–2.21	0.05

**Table 3 jcm-12-07596-t003:** Patients’ preferences with respect to long life vs. quality of life by gender, age groups, and multimorbidity.

	Males			Females			Total		
	OR	95% CI	*p*-Value	OR	95% CI	*p*-Value	OR	95% CI	*p*-Value
Age below 60	1.00			1.00			1.00		
Age 60–69	1.34	0.91–1.97	0.14	1.80	0.90–3.60	0.10	1.43	1.02–2.01	0.04
Age 70–79	1.16	0.80–1.66	0.44	2.46	1.29–4.70	0.01	1.42	1.03–1.94	0.03
Age above 80	1.19	0.72–1.97	0.50	3.98	1.73–9.14	<0.01	1.72	1.13–2.64	0.01
Multimorbidity	0.80	0.60–1.09	0.16	0.67	0.41–1.09	0.10	0.78	0.60–1.00	0.05

## Data Availability

Due to ethical reasons, data can only be transferred after an application and upon request to the ethical commission of the Medical Faculty of the Martin-Luther University Halle-Wittenberg.

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
