# Peer review of "Living Longer or Better—Patient’s Choice in Cardiac Surgery Is Gender-Dependent—A Multicenter Study"

_jcm, 2023, doi:10.3390/jcm12247596_

Round 1
Reviewer 1 Report
Comments and Suggestions for Authors
Thank you very much for inviting me to review the article entitled "Living longer or better – patient's choice in cardiac surgery is gender-dependent – a multicenter study”.
This study aims to obtain information on cardiac surgery patient preferences regarding quality of life (QoL) or lifespan (LoL). The current study is interesting and addresses an important issue among cardiac surgery patients whose age is steadily increasing. The study shows the difference in preference between quality or length of life according to gender and age.
1. Please indicate how dementia was assessed and the exclusion criteria related to dementia.
From my perspective, there is a lack of attempt to explain this phenomenon.
2. From the above analysis, the question arises as to whether there was an association between preferences and co-morbidities or multimorbidity (defined as 2 disease entities in an individual)?
3. Is it perhaps due to education level, income, social factors (family, children)?
A study worth considering after attempting a deeper analysis of the phenomenon.
Author Response
We sincerely appreciate given comments and suggestions, some of them identified substantial gaps and problematic points in the manuscript, all of which we have tried to address.
Reviewer 1:
Thank you very much for inviting me to review the article entitled "Living longer or better – patient's choice in cardiac surgery is gender-dependent – a multicenter study”.
This study aims to obtain information on cardiac surgery patient preferences regarding quality of life (QoL) or lifespan (LoL). The current study is interesting and addresses an important issue among cardiac surgery patients whose age is steadily increasing. The study shows the difference in preference between quality or length of life according to gender and age.
- Please indicate how dementia was assessed and the exclusion criteria related to dementia.
From my perspective, there is a lack of attempt to explain this phenomenon.
The patients with severe dementia will not undergo heart surgery but instead get an intervention by the cardiologists. These patients do not appear within the cardiac surgery clinics. In addition, within the pre-treatment consultation, the interviewer tests the understanding of the consultation / treatment. Therefore, all patients with dementia are excluded. The methods section is now accordingly modified.
- From the above analysis, the question arises as to whether there was an association between preferences and co-morbidities or multimorbidity (defined as 2 disease entities in an individual)?
We thank the reviewer for this suggestion. We have calculated the co-morbidities as well as multimorbidity (more than two diseases/co-morbidities). As shown in Table 3 (new table in the manuscript), we do not see a change in the results. We therefore can exclude that these factors really influence the patient preferences.
- Is it perhaps due to education level, income, social factors (family, children)?
We asked the patients for their familiar situation (living alone, in a family, or a nursing home).
There is no association between the patient's preferences and the familial situation.
Reviewer 2 Report
Comments and Suggestions for Authors
Dear Authors,
Firstly, I would like to congratulate you on writing this interesting and well-designed scientific article. Furthermore, I understand that the article presents many results that corroborate the hypothesis discussed by the authors.
I would like you to please clarify some points that I consider important and decisive at the time the patient participated in the study:
1) Was it assessed whether the patients had or had psychiatric disorders such as, for example, depression or anxiety disorders in the past or at the time of the research?
2) Was it assessed whether the patients had or had neurodegenerative diseases in the past or at the time of the research?
3) Was it assessed whether the patients had or were suffering from oncological diseases in the past or at the time of the research?
4) Was the patients’ self-esteem assessed?
I understand that such answers directly influence the answers provided by the interviewed patients, since such situations may be more relevant, in some cases, than which gender the patient belongs to. Furthermore, you discussed the differences in the expression of the COMT enzyme, while published data also demonstrate differences in the expression of the MAO enzyme, which are directly involved in neurodegenerative diseases and psychiatric disorders, which is why I believe that the questions above could influence the answers provided. by the interviewed patients.
Kind regards
Comments on the Quality of English LanguageI think the quality of the English language is excellent.
Author Response
Dear Authors,
Firstly, I would like to congratulate you on writing this interesting and well-designed scientific article. Furthermore, I understand that the article presents many results that corroborate the hypothesis discussed by the authors.
We thank the reviewer for the positive overall evaluation.
I would like you to please clarify some points that I consider important and decisive at the time the patient participated in the study:
1) Was it assessed whether the patients had or had psychiatric disorders such as, for example, depression or anxiety disorders in the past or at the time of the research?
The study design was not focused on psychiatric disorders. Instead, the study was focused on typical co-morbidities like diabetes, arrhythmia, COPD, or kidney diseases.
2) Was it assessed whether the patients had or had neurodegenerative diseases in the past or at the time of the research?
As mentioned above, this was not assessed. Patients with major neurodegenerative diseases will not undergo heart surgery but instead, get intervention by the cardiologists. On the other hand, we see the problem and address it in the discussion section.
3) Was it assessed whether the patients had or were suffering from oncological diseases in the past or at the time of the research?
None of the patients had a prognosis relevant malign disease nor had treatments like chemotherapies.
4) Was the patients’ self-esteem assessed?
As most of the involved clinics do not have experts / psychologically trained coworkers, we did not include assessments like the patients' self-esteem.
The reviewer mention interesting assessments that should be included in a following study.
Round 2
Reviewer 2 Report
Comments and Suggestions for Authors
Dear Authors,
After reading the article, I realized that you made all the adjustments requested during the first review of the work and, therefore, I understand that the current version submitted to the magazine is compliant.
Congratulations on the great work done.
Kind regards,
Comments on the Quality of English LanguageThe English used in writing the article is easy to read and technically adequate.
Author Response
Thank you very much for your comments.